# Synthesis, Antimicrobial, and Antibiofilm Activities of Some Novel 7-Methoxyquinoline Derivatives Bearing Sulfonamide Moiety against Urinary Tract Infection-Causing Pathogenic Microbes

**DOI:** 10.3390/ijms24108933

**Published:** 2023-05-18

**Authors:** Mostafa M. Ghorab, Aiten M. Soliman, Gharieb S. El-Sayyad, Maged S. Abdel-Kader, Ahmed I. El-Batal

**Affiliations:** 1Drug Chemistry Laboratory, Drug Radiation Research Department, National Center for Radiation Research and Technology (NCRRT), Egyptian Atomic Energy Authority (EAEA), Cairo 11787, Egypt; mmsghorab@yahoo.com (M.M.G.); aiten_mahmoud@yahoo.com (A.M.S.); 2Drug Microbiology Laboratory, Drug Radiation Research Department, National Center for Radiation Research and Technology (NCRRT), Egyptian Atomic Energy Authority (EAEA), Cairo 11787, Egypt; gharieb.elsayyad2017@gmail.com (G.S.E.-S.); aelbatal2000@gmail.com (A.I.E.-B.); 3Department of Pharmacognosy, College of Pharmacy, Prince Sattam Bin Abdulaziz University, Al-Kharj 11942, Saudi Arabia; 4Department of Pharmacognosy, College of Pharmacy, Alexandria University, Alexandria 21215, Egypt

**Keywords:** quinoline, sulfonamide, antimicrobial activity, urinary tract infection, reaction mechanism

## Abstract

A new series of 4-((7-methoxyquinolin-4-yl) amino)-*N*-(substituted) benzenesulfonamide **3(a–s)** was synthesized via the reaction of 4-chloro-7-methoxyquinoline **1** with various sulfa drugs. The structural elucidation was verified based on spectroscopic data analysis. All the target compounds were screened for their antimicrobial activity against Gram-positive bacteria, Gram-negative bacteria, and unicellular fungi. The results revealed that compound **3l** has the highest effect on most tested bacterial and unicellular fungal strains. The highest effect of compound **3l** was observed against *E. coli* and *C. albicans* with MIC = 7.812 and 31.125 µg/mL, respectively. Compounds **3c** and **3d** showed broad-spectrum antimicrobial activity, but the activity was lower than that of **3l**. The antibiofilm activity of compound **3l** was measured against different pathogenic microbes isolated from the urinary tract. Compound **3l** could achieve biofilm extension at its adhesion strength. After adding 10.0 µg/mL of compound **3l**, the highest percentage was 94.60% for *E. coli,* 91.74% for *P. aeruginosa,* and 98.03% for *C. neoformans*. Moreover, in the protein leakage assay, the quantity of cellular protein discharged from *E. coli* was 180.25 µg/mL after treatment with 1.0 mg/mL of compound **3l**, which explains the creation of holes in the cell membrane of *E. coli* and proves compound **3l**’s antibacterial and antibiofilm properties. Additionally, in silico ADME prediction analyses of compounds **3c**, **3d,** and **3l** revealed promising results, indicating the presence of drug-like properties.

## 1. Introduction

The overuse of antibiotics results in antimicrobial resistance (AMR), a growing global health concern [1]. This causes the emergence of bacterial strains that are resistant to antibiotics, causing infections which are more challenging to treat and raising the possibility of the spread of disease [2]. Thus, the number of multi-drug resistant (MDR) bacterial strains has grown since the 1960s [3]. Although there are many efficient antimicrobial medications available in the clinic, spontaneous genetic changes occurring in bacteria can influence the effectiveness of these drugs. The excessive and inappropriate use of antimicrobial medications promotes these genetic changes [4], therefore, antimicrobial drugs can become less effective in a shorter period of time, causing the rapid development of resistance [5]. However, due to rapidly evolving resistance, infectious diseases continue to pose one of the greatest risks to public health [6]. In fact, antibiotic resistance to bacterial infectious diseases is thought to be the cause of a significant number of annual fatalities [7]. Methicillin-resistant *S. aureus* (MRSA), vancomycin-resistant *S. aureus* (VRSA), and quinolone-resistant *S. aureus* (QRSA) strains are just a few examples of the drug-resistant bacteria that are on the increase globally [8,9]. Additionally, the pathogens known as the ESKAPE set of multidrug resistant (MDR) strains are regarded as international targets [10].

Urinary tract infection (UTI) is a prevalent condition that can be classified as either uncomplicated or complicated [11]. Uncomplicated UTIs can affect healthy individuals with normal urinary tracts and are typically caused by uropathogenic *Escherichia coli* (UPEC). Conversely, complicated UTIs can occur in patients with abnormal urinary tracts or those who are immunocompromised [12]. In the case of complicated UTIs, a broad range of pathogens can be involved, and treatment with antibiotics may be less effective, leading to a higher incidence of relapse. The rapid identification of the pathogen’s resistance profile is crucial in diagnosing and treating UTIs [13,14].

Microbial biofilms have an impact on several diseases, and the characteristics of these microorganisms that are associated with biofilms can result in significant antibiotic resistance. The biofilm matrix, which functions as a mechanical barrier, may interfere with immune response agents and antibiotic therapies [15]. A deficiency in nutrients or a non-growing, yet hardy, phenotype that enables microbes to endure environmental stressors, such as exposure to antibiotics, may also cause bacteria to develop high levels of antibiotic resistance. This issue calls for prompt action and highlights the need to explore innovative ways to create new, effective, and safe antimicrobial drugs [16].

Certain strains of *E. coli* are responsible for increased morbidity and mortality, particularly in immunocompromised patients using various medical devices such as urethral and intravascular catheters [17]. *E. coli*-triggered infections are challenging to treat due to biofilm formation [18]. These biofilms are made of bacterial colonies surrounded by a matrix of extracellular polymeric substances (EPS) which shields the microbes from adverse environmental conditions leading to infection. Besides being responsible for recurrent urinary tract infections, *E. coli* biofilm is the cause of innate medical device-related infectivity [19]. Biofilm reduces the diffusion of conventional antibiotics and renders the cells resistant to antibiotics [20]. *E. coli* can become resistant by altering the target enzymes, reducing the permeability of the cell to inhibit their entry, or actively pumping the drug out of the cell [21]. All these resistance mechanisms can play a role in antibiotic resistance; however, target site mutations appear to be the most important mechanism [22]. Biofilm is considered an important target in the fight against drug-resistant bacterial infections, suggesting an urgent need to explore alternative therapeutic agents [23].

Quinolines constitute an important class of compounds due to their resemblance to ciprofloxacin, which treats various bacterial infections, including bone, joint, and intra-abdominal infections [24]. Several drugs that fight cancer [25], parasites [26], tuberculosis [27], malaria [28], and viruses such as SARS-CoV-2 [29,30] have a quinoline backbone. Their high activity and limited toxicity make them better treatment candidates and the drug of choice in various cases. Many synthetic and naturally occurring quinolines, such as quinine, ciprofloxacin, and hydroxychloroquine, were reported to have antimicrobial activity [31,32], as shown in Figure 1. Quinolines are antibacterial candidates involved in DNA replication, transcription, and recombination in bacterial cells though the inhibition of topoisomerase II (DNA gyrase) and topoisomerase IV [33,34]. The blockage of these enzymes is an essential target for discovering and developing new antibacterial drugs [35].

Quinolones are widely used to treat urinary tract and respiratory infections [36]. Because of their common use and overuse, several quinolone-resistant bacterial strains have emerged since the 1990s. Like other antibacterial agents, the increase in quinolone resistance threatens the clinical utility of this important drug class. The 4-quinolones were introduced for medical use in 1964 [37]. The quinolone antibiotics are active against a wide range of Gram-negative bacteria, with minimal inhibitory concentrations (MICs) in the nanomolar range, and are relatively potent towards many Gram-positive bacteria [38]. Their activity is due to the inhibition of DNA replication though the inhibition of DNA gyrase and topoisomerase IV activities to varying degrees, depending on the pathogen [39].

On the other hand, sulfonamides were the first agents discovered to be active against pyogenic bacterial infections [40]. Additionally, the chemotherapeutic action of sulfonamides has been the subject of extensive research for many years [41,42,43,44]. The antibacterial activity of sulfonamides was proved to be due to the competitive inhibition of dihydropteroate synthase (DHPS), which is crucial for folate synthesis, as it consequently inhibits DNA replication [45,46]. Sulfonamides were found to exhibit broad-spectrum activity against Gram-positive and Gram-negative bacterial strains. The use of sulfonamides has lately been reduced, owing to the development of allergic reaction conditions [47,48]. However, they are still used though a hybridization strategy to develop new agents with higher antibacterial potential [43,49,50].

As a result of the literature review, some new quinoline derivatives that exhibited excellent antibacterial activity were used to design our target compounds, as shown in Figure 2. Bazin et al. [51] reported the diethyl ((*N*-(4-bromophenyl) sulfamoyl) (2-chloro-8-methylquinolin-3-yl) methyl) phosphonate **A** to have a very potent activity against *E. coli*, with an MIC of 0.125 μg/mL. Moreover, the quinoline benzodioxole derivative **B** showed excellent antibacterial activity, with an MIC of 3.125 μg/mL against *E. coli* and *S. aureus* strains [52]. The quinoline-3-carbonitrile derivative **C** synthesized by Khan et al. [53] exhibited antibacterial potential against Gram-negative bacteria with the highest activity towards *E. coli*, with an MIC of 4 μg/mL.

In continuation of our research aimed at discovering new antimicrobial agents with improved activity [54], we herein describe the design and synthesis of a set of compounds targeting UTI infections. The current study employed a hybridization strategy to evaluate the antibiofilm effects of the quinolone scaffold and the sulfonamide moiety by changing the sulfonamide-privileged pharmacophore, which could increase the compound’s efficacy (Figure 2). The structures of these target derivatives were confirmed. Consequently, the antimicrobial effects of all the compounds were screened against various Gram-positive bacteria, Gram-negative bacteria, and fungi. The antibiofilm potential of the most potent compound was also investigated.

## 2. Results and Discussion

### 2.1. Chemistry

This work aims to design and synthesize a new series of 4-((7-methoxyquinolin-4-yl) amino)-*N*-(substituted) benzenesulfonamide **3(a–s)** to be evaluated as antimicrobial and antibiofilm agents against pathogenic microbes.

The reaction of 4-chloro-7-methoxyquinoline **1** with a series of sulfonamides **2(a–s)** in dimethylformamide (DMF) under reflux afforded compounds **3(a–s)**. The structures of these compounds were confirmed though spectral and elemental analysis. The IR spectra of compounds **3(a–s)** showed absorption bands for NH, CH aromatic, CH aliphatic, and SO_2_ at their specified regions. The ^1^H-NMR spectra of **3(a–s)** exhibited a singlet in the range of 3.78–3.86 ppm corresponding to the OCH_3_, a singlet at 8.65–11.25 ppm for the SO_2_NH, and aromatic hydrogens in the aromatic region. The ^13^C-NMR spectra of **3(a–s)** exhibited signals in the range of 54.18–56.53 ppm assigned to the OCH_3_ and 151.17–153.47 ppm for the CN, respectively. The IR spectrum of **3b** indicated the presence of the COCH_3_ band at 1678 cm^−1^. The isoxazole derivative **3f** exhibited one CH_3_ that appeared at 2.51 and 13.21 ppm in ^1^H-NMR and ^13^C-NMR, respectively. The ^1^H-NMR spectrum of **3h** showed a singlet at 2.54 ppm and a signal at 23.81 ppm in ^13^C-NMR, attributed to the CH_3_, while the oxazole derivative **3i** exhibited two singlet signals in ^1^H NMR at 2.11 and 2.28 ppm and two signals in ^13^C NMR at 31.20 and 36.23 ppm, corresponding to the 2CH_3_ groups. The ^1^H-NMR spectrum of **3j** revealed a CH_3_ singlet at 2.65 ppm and a signal at 16.46 ppm in ^13^C-NMR. The ^1^H-NMR spectra of **3k** and **3l** revealed singlets of 2CH_3_ at 2.44, 2.61, and 2.40 ppm, while ^13^C-NMR of **3k** and **3l** showed signals at 16.35, 16.78 ppm, and 26.52 ppm for the 2CH_3_. The ^1^H-NMR of **3m** and **3n** demonstrated OCH_3_ at 3.85 and 3.79 ppm, while the ^13^C-NMR of **3m** and **3n** displayed signals at 56.63 and 55.74 ppm for the OCH_3_. The ^1^H-NMR of **3q** and **3r** displayed 2OCH_3_ groups at 3.78, 3.75, and 3.86 ppm, respectively, while the ^13^C-NMR of **3q** and **3r** showed two signals at 54.89, 56.17, 55.95, and 56.48 ppm for the 2OCH_3_ (Figure 1).

### 2.2. Antimicrobial Activity of the Newly Synthesized Compounds

Recently, quinoline derivatives have been widely used to treat resistant microbes that produce slim biofilms [55,56]. Therefore, all the prepared compounds were screened for their antimicrobial potential; among them, compounds **3c**, **3d**, and **3l** exhibited the highest activity. The antimicrobial activity of compounds **3c**, **3d**, and **3l** against different bacterial and fungal strains was evaluated, as shown in Table 1 and Figure 3. Overall, all the designed compounds exhibited promising antimicrobial activity against all the tested bacterial and fungal strains compared to AMC/Nyst as conventional antimicrobial agents. Compounds **3c**, **3d,** and **3l** were significantly more active than AMC/Nyst.

The results revealed that compound **3l**, bearing sulfamethazine, has the highest effect on most of the tested bacterial and fungal strains, when compared to the other sulfaguanidine derivative **3c** and the sulfapyridine derivative **3d**. Table 1 illustrates that compound **3l** (0.1 mg/mL) had the highest impact on *E. coli* among all tested bacterial strains. The inhibition zone was 21 mm. Additionally, it displayed the most potent activity against *C. albicans* among all the tested fungal strains, with an inhibition zone of 18 mm. Moreover, the sulfamethazine derivative **3l** at a concentration of 0.1 mg/mL showed promising antimicrobial activity against *E. coli, P. aeruginosa, S. aureus, B. subtilis, C. albicans,* and *C. neoformans,* with inhibition zones of 21.0, 16.2, 18.0, 16.0, 18.0, and 10.0 mm, respectively. Furthermore, compounds **3c** and **3d** showed antimicrobial activity, but this activity was lower than that of compound **3l**; the highest effect was observed against *E. coli* and *C. albicans*, with inhibition zones of 18.8 and 15.0 mm, respectively, for compound **3d** and 12.0 and 11.2 mm for compound **3c**. This was in agreement with the work of Tailor et al. [57], which indicates the high antimicrobial activity of sulfamethazine against *S. aureus* and *E. coli* strains. Moreover, Ragab et al. [58] and Chen et al. [59] indicated the promising effect of sulfaguanidine and sulfapyridine derivatives against *E. coli, P. aeruginosa, S. aureus, B. subtilis,* and *C. albicans*, respectively.

Additionally, the MICs of all tested samples (compounds **3c**, **3d**, and **3l**) were determined, as shown in Table 1. The results showed that among the other microbial strains examined, compounds **3l**, **3d,** and **3c** exhibited the best MICs, with values between 7.812 and 500 µg/mL, towards the tested bacteria and unicellular fungi. Additionally, *E. coli* was found to be the most susceptible of the tested bacteria, with the MICs of compounds **3l**, **3d**, and **3c** of 7.81, 31.25, and 62.50 µg/mL, respectively. However, the MIC of all the compounds against *E. coli, P. aeruginosa*, and *S. aureus* ranged from 125 to 500 µg/mL, which was lower than that of *B. subtilis*. Eventually, the designed compounds displayed good antimicrobial activity against bacteria, as well as unicellular and multicellular fungi when compared to the activity of traditional antimicrobial agents (AMC/Nyst).

### 2.3. Structure Activity Relationship (SAR) Study

*Regarding the open chain derivatives* **3(a–c)**:

The open chain derivatives **3(a–c)** seemed to have lower activity compared to the aromatic heterocyclic derivatives **3(d–s)**, except for the guanidino derivative **3c**. Moreover, increasing the length of the side chain seems to boost antimicrobial activity. The sulfanilamide derivative **3a** exhibited the least potent activity among all the synthesized compounds. The introduction of an acetyl group to the sulfanilamide, as in **3b**, enhanced its activity against *E. coli* and *S. aureus*. The replacement of the amino group in **3a** with a guanidino group in **3c** led to a significant increase in activity against *E. coli,* with a relative increase in activity against *P. aeruginosa*, *S. aureus*, and *C. albicans*.

*Regarding the heterocyclic aromatic derivatives* **3(d–s)**:

The 6-membered heterocyclic ring derivatives displayed enhanced activity compared to the 5-membered and fused heterocyclic derivatives.

*Regarding the 6-membered heterocyclic derivatives* **3d**, **3e**, **3h**, **3k**, **3l**, **3m**, **3n**, **3q**, *and* **3r**:

Introducing a terminal hydrophobic 6-membered heterocyclic ring was found to enhance antimicrobial activity. It is apparent from the results that the 2-pyrimidinyl derivatives are more potent than the 4-pyrimidinyl derivatives.

*The 2-pyrimidinyl derivatives* **3e**, **3h**, **3l**, *and* **3m**:

The unsubstituted 2-pyrimidinyl **3e** displayed moderate activity towards *E. coli, S. aureus,* and *C. albicans*. The introduction of a monomethyl group at the 4-position, as in **3h**, resulted in a decline in activity towards *C. albicans*, while retaining the same activity towards *E. coli* and *S. aureus*. The replacement of the methyl group in **3h** with a more electron-donating methoxy group at the 6-position, as in **3m**, diminished the activity towards *S. aureus*, while displaying moderate activity towards *P. aeruginosa*. Additionally, the introduction of dimethyl groups in positions 4 and 6 (as in **3l**) greatly enhanced the activity towards all the tested strains, with the highest potency towards *E. coli,* followed by *S. aureus* and *C. albicans*, while retaining moderate activity towards *P. aeruginosa* and *B. subtilis.* Compound **3l** was the most potent in this series, proving that increasing the number of hydrophobic groups attached to the pyrimidine ring is favorable. The replacement of the unsubstituted pyrimidine ring (as in **3e**) with an unsubstituted pyridine (as in **3d**) is also favorable, as it enhanced the activity against *E. coli, P. aeruginosa, B. subtilis*, and *C. albicans*.

*The 4-pyrimidinyl derivatives* **3k**, **3n**, *and* **3q:**

The introduction of dimethoxy groups in positions 2 and 6, as in **3q**, displayed increased activity compared to that of the monomethoxy group, as in **3n**. These dimethoxy groups increased hydrophobicity and improved the activity towards *S. aureus* and *C. albicans* while retaining the same activity towards *E. coli.* On the other hand, the replacement of the dimethoxy groups in **3q** with the dimethyl groups in **3k** diminished the activity. Thus, di-substitution is more favorable than mono-substitution, and the presence of stronger activating groups (methoxy) is favorable and enhances the activity towards *S. aureus* and *C. albicans.*

*The 5-membered ring derivatives* **3f**, **3g**, **3i**, **3j**, *and* **3s**:

The 5-methoxazol-3yl derivative **3f** showed enhanced activity towards *E. coli* compared to the 4,5-dimethyloxazol-2-yl derivative **3i.** The introduction of dimethyl groups to the oxazole did not enhance the activity. On the other hand, the replacement of the thiazole in **3g** with thiadiazole, bearing a methyl group at the 5-position (as in **3j**), enhanced the activity against *E. coli* and *C. albicans* from poor to moderate. The 1-phenyl-pyrazol-5-yl derivative **3s** showed moderate activity towards *E. coli, P. aeruginosa,* and *C. albicans*. Thus, increasing the number of nitrogen atoms inside a 5-membered heterocyclic ring can enhance the activity against *E. coli* and *C. albicans,* specifically.

*Regarding the fused heterocyclic derivatives* **3o** *and* **3p**:

The introduction of a bulky bicyclic structure consisting of two fused 6-membered aromatic rings, as in **3p**, or a 5-membered ring fused to benzene, as in **3o,** can greatly affect their activity. The indazole derivative **3o** showed improved activity compared to the quinoline derivative **3p** against *P. aeruginosa, S. aureus,* and *C. neoformans*, proving that the 5-membered fused ring is more favorable than the 6-membered fused ring derivatives (Figure 4) (see Appendix A).

### 2.4. Antibiofilm Potential of Compound **3l**

Exo-polysaccharide can be used to detect the biofilm development of deadly bacteria [60]. The antibiofilm behavior of the integrated compound towards various pathogenic bacteria and unicellular fungi was defined using the tube design [61].

The experimental data pointed to the antibiofilm activity of compound **3l** (the most potent compound) against *E. coli* (an example of a sensitive pathogenic bacteria). The complete results are: (I) the regular microbial growth and reproduction of the distinguished ring in the absence of the integrated compound **3l** and the restraint of the microbial growth in the presence of compound **3l**, (II) the possibility of staining of the established biofilm with Crystal Violet (CV), which is a qualitative measurement system, and (III) the removal and separation of the adhered microbial cells following an ethanol reaction for the semi-quantitative evaluation of the biofilm interruption percentage (Table 2).

The results showed the tube design to determine the antibiofilm potential of compound **3l** against *E. coli*, which created a thick whitish-yellow layer at the air–liquid interface in the solution of compound **3l**. The produced matte layers were fully adhered across the walls of the designed tubes and developed a blue color following the staining with CV. Next, a dark blue color was created in the produced solution, subsequently dissolving CV with absolute ethanol.

Additionally, a remarkable negative impact was noted, as the cells of the tested bacteria do not produce biofilm layers, and the ring construction was blocked in the tubes containing the *E. coli* cells and compound **3l** (10 µg/mL). Moreover, the adherent cell color was soft, and the blue color was faintly developed following the ethanol addition.

A UV-Visible spectrophotometer examined the semi-quantitative measurement of the repression percentage (%). The optical density (O.D.) was measured at 570 nm following the termination of the CV-stained biofilms, which was recognized as a result of their production [61].

Table 2 illustrates the inhibition % after adding 10.0 µg/mL of compound **3l**, showing that the highest percentage of *E. coli* was 94.60%, the highest percentage of *P. aeruginosa* was 91.74%, and the highest percentage of *C. neoformans* was 98.03%. Note that compound **3l** achieved the biofilm extension at its adhesion strength, which is the initial starting level in the antimicrobial method [62]. The difference in the hindrance percentage may be linked to several constituents, such as the significant potential of the antimicrobial factors to be connected to the surface due to the enhanced surface area of the integrated compound **3l** and its particle size, as well as the attack mode and various chemical properties affecting the association and interaction of compound **3l** among biofilm-producing bacteria [61,63,64]. Figure 5 presents a diagram showing the antibiofilm activity of compound **3l** (as inhibition %) towards various pathogenic microbes.

### 2.5. Kinetics of E. coli Growth (Growth Curve)

The impact of compound **3l** on *E. coli* growth kinetics was investigated. As shown in Figure 6, the control sample’s *E. coli* development rate appears rapid. The O.D. of the control sample at λ = 600 nm was 2.18. In contrast, the OD_600_ values of the compound **3l**-treated cells were lower than those of the control sample due to the exceptional antibacterial action of compound **3l**.

The bacterial growth inhibition rate resulting from compound **3l** treatment started from the first time of observation until the endpoint at 24 h (O.D. 1.01). There was no notable difference between the influences of compound **3l** concentrations at the beginning of observation. Additionally, compound **3l** exerted an extra inhibitory impact compared to the control, as established by the O.D. results (Figure 6). The effects showed that the *E. coli* growth rates without compound **3l** were greater than the growth rates with compound **3l**. For a compound to have the antimicrobial potential to kill pathogenic microbes, it must adhere to its target locations on the microbial cells and settle in a precise number of critical areas connected to its concentration within the pathogenic microbes.

### 2.6. Determination of Protein Leakage from Bacterial Cell Membranes

The quantities of protein discharged in the suspension of the treated *E. coli* cells were determined by applying the Bradford method [65]. From Figure 7, it can be seen that the quantity of cellular protein discharged from *E. coli* is directly proportional to the concentration of compound **3l,** which is found to be 180.25 µg/mL after the treatment with 1.0 mg/mL of compound **3l**. This result proves the antibacterial characteristics of the synthesized compound **3l**, and explains the creation of holes in the cell membrane of *E. coli*, which produced the oozing out of the proteins from the *E. coli* cytoplasm.

These test outcomes revealed that compound **3l** improved the permeability of *E. coli* cell membranes; therefore, it could be assumed that the confusion of membranous permeability would be a vital portion of the repression of bacterial mass. Related studies, such as those in [66,67], described comparable outcomes when incorporating ferrites, which revealed concentration-dependent destabilization in the cell membrane of bacterial cells and pointed to leakage of their intracellular substance into the extracellular form (bacterial cell suspension).

Paul et al. [68] proved that the difference in bacterial cell membrane permeability was shown in the percentage difference in the corresponding electric conductivity. It was reported that the percentage of relative electric conductivities of tested samples improved with the increase in the concentration of the treated compounds. The integrity of the bacterial cell membrane was defined by the analysis of the discharge of cell components of the bacteria, such as proteins; the leakage developed with time, as there was constant cell membrane injury that pointed to the leakage of cell components driving cell destruction.

### 2.7. Reaction Mechanism Determination by SEM

SEM analysis was conducted to demonstrate the possible antimicrobial mechanism against *E. coli*, as noted in Figure 8. The SEM study regarding the control bacterial cells in the absence of compound **3l** presented bacterial groups that typically extended and grew with a regular surface and a normal shape and count, as displayed in Figure 8a.

Following compound **3l** treatment, unusual morphological irregularities were identified in *E. coli* (Figure 8b), including the semi-lysis of the outer surface in some bacterial cells established by deformations of the *E. coli* cells. On the other hand, the synthesized compound **3l** achieved complete lysis of the bacterial cell, as well as cell malformation, decreasing the total viable number (Figure 8b), and creating holes on the surface of bacterial cells. A white layer was formed over the bacterial cells due to the chemisorption attractions between the lone pairs of electrons found in the active site in compound **3l** and the bacterial cell wall, which was confirmed by the membrane leakage assay.

On the other hand, El-Sayyad et al. [69] discussed the antibacterial reaction mechanism after conducting SEM imaging against *E. coli* treated with the synthesized Se NPs-gentamicin (CN) nano-drug and found that the *E. coli* cells showed morphological modifications after the treatment with Se NPs-CN. A noticeable elevation in the hardness of the bacterial cell surface and bacterial cell malformation suggested that it was suppressed and regulated by Se NPs-CN. They were also reduced to a viable count, and the biofilm was hindered.

### 2.8. In Silico ADME Study

The success of a compound for therapeutic usage depends on many factors, including absorption, distribution, metabolism, and excretion (ADME). The bioavailability of drugs is highly influenced by physicochemical factors. One of the most crucial aspects of drug development is the prediction of those features before experimental studies. The optimization of the pharmacokinetics for new drugs involves the investigation of ADME features in a dynamic way. The SwissADME online tool [70] was used to assess the physicochemical properties of the most active compounds, **3c**, **3d**, and **3l**. Topological polar surface area (TPSA) shows how easily compounds can cross the blood–brain barrier and be absorbed in the intestine. The drug must have a TPSA value of less than 90 in order to cross the blood–brain barrier. The number of flexible bonds also has a significant impact on how molecules interact with one another and attach to the binding sites, with the majority of synthetic compounds having a high number of flexible bonds. HBD and HDA effectively demonstrated the suitability of the novel target compounds as possible therapies, according to the Lipinski rule of five (RO5) [71]. The results of the ADME predictions in Table 3 demonstrated molecular weight (Mol. Wt.), the logarithm of the partition coefficient (log P), gastrointestinal (GI) absorption, the CYP1A2/CYP3A4 substrate, and Lipinski’s rule of five. A drug must have a high rate of gastrointestinal absorption to be orally active, and compound **3d** demonstrated a high rate of GI absorption. The most important factor influencing absorption is bioavailability, which measures the amount of the drug in the bloodstream. It is interesting to note that compounds **3c**, **3d**, and **3l** have high bioavailability scores. The target compounds were discovered to be skin permeable. Additionally, none of the compounds that were synthesized violated the Lipinski rule of 5. Last but not least, they showed no PAINS (pan-assay interference compounds) alerts. The results of ADME demonstrated that the most active compounds, **3c**, **3d**, and **3l**, possess drug-like properties.

## 3. Materials and Methods

Thin layer chromatography was performed on pre-coated silica gel plates (Kiesel gel 0.25 mm, 60 G F 254, Merck, Munich, Germany), and the solvent system used was chloroform/methanol (7:3). The spots were detected under ultraviolet light. The melting points were measured (uncorrected) using a melting point apparatus (Sanyo Gallen Kamp, Cambridge, UK). IR spectra were obtained using an FT-IR spectrophotometer (Perkin Elmer, Massachusetts, USA). NMR spectra were acquired in DMSO-*d_6_* using an NMR spectrophotometer (Bruker AXS Inc., Zurich, Switzerland) operating at 500 MHz for ^1^H and 125.76 MHz for ^13^CNMR. The chemical shifts were reported in δ values (ppm) relative to tetramethylsilane as the internal standard. Mass spectra were run on the direct inlet part of the mass analyzer in Thermo Scientific GCMS model ISQ LT (Massachusetts, USA). The elemental analyses were conducted on a model 2400 CHNSO analyzer (Perkin Elmer, Massachusetts, USA). The starting material, 4-chloro-7-methoxyquinoline **1**, and the sulfonamide derivatives were obtained from Sigma-Aldrich.

### 3.1. Chemistry

#### 3.1.1. General Procedure for the Synthesis of Compounds **3(a–s)**

A mixture of 4-chloro-7-methoxyquinoline **1** (1.93 g, 0.01 mol) and sulfonamide derivatives **2(a–s)** (0.01 mol) was refluxed in dimethylformamide (20 mL) for 24 h. The obtained solid was crystallized from ethanol to give **3(a–s)**.

#### 3.1.2. 4-((7-Methoxyquinolin-4-yl)amino)benzenesulfonamide **(3a)**

Yield 54%, m.p. 240–242 °C. IR: 3387, 3321, 3265 (NH, NH_2_), 3011 (arom.), 2951, 2868 (aliph.), 1632 (CN), 1376, 1148 (SO_2_). ^1^H-NMR δ: 3.15 (s, 2H, NH_2_), 3.85 (s, 3H, OCH_3_), 7.09 (d, 1H, *J* = 8 Hz), 7.20 (s, 1H), 7.33–7.40 (m, 3H), 7.76 (d, 2H, *J =* 9 Hz, AB), 8.15 (d, 1H, *J =* 6 Hz), 8.57 (d, 1H, *J =* 8 Hz), 9.20 (s, 1H, NH). ^13^C NMR δ: 55.85, 103.25, 107.94, 117.73 (2), 120.06 (2), 124.25, 127.73 (2), 137.73, 144.98, 146.73, 151.17(2), 160.71. MS (*m*/*z*, RI %): 329.52 (M^+^) (51.12), 330.86 (M + 1) (41.90), 44.20 (100). Anal. Calcd. for C_16_H_15_N3O_3S_ (329.08): C, 58.34; H, 4.59; N, 12.76. Found: C, 57.98; H, 4.45; N, 12.38.

#### 3.1.3. N-((4-((7-Methoxyquinolin-4-yl)amino)phenyl)sulfonyl)acetamide **(3b)**

Yield 76%, m.p. >300 °C. IR: 3364, 3236 (2NH), 3068 (arom.), 2979, 2870 (aliph.), 1678 (CO), 1626 (CN), 1343, 1160 (SO_2_). ^1^H-NMR δ: 2.00 (s, 3H, COCH_3_), 3.86 (s, 3H, OCH_3_), 6.98 (d, 1H, *J* = 7 Hz), 7.30–7.46 (m, 4H), 7.79 (d, 2H, *J =* 7.5 Hz, AB), 8.05 (d, 1H, *J =* 6.5 Hz), 8.40 (d, 1H, *J =* 7.5 Hz), 8.90 (s, 1H, NH), 11.25 (s, 1H, SO_2_NH). ^13^C NMR δ: 23.77, 56.53, 100.27, 100.68, 112.68 (2), 116.32, 117.31, 119.08, 126.47, 129.77 (2), 143.01, 148.30, 149.40 (2), 153.97, 169.47. Anal. Calcd. for C_18_H_17_N_3_O_4_S (371.09): C, 58.21; H, 4.61; N, 11.31. Found: C, 58.55; H, 4.91; N, 11.70.

#### 3.1.4. N-(Diaminomethylene)-4-((7-methoxyquinolin-4-yl)amino)benzenesulfonamide **(3c)**

Yield 80%, m.p. > 300 °C. IR: 3380, 3342, 3265 (NH, NH_2_), 3058 (arom.), 2958, 2810 (aliph.), 1621 (CN), 1335, 1162 (SO_2_). ^1^H-NMR δ: 3.85 (s, 3H, OCH_3_), 6.52 (d, 1H, *J* = 8.5 Hz), 7.55 (d, 1H, *J* = 2 Hz), 7.55–7.63 (m, 5H), 7.90 (d, 1H, *J =* 8 Hz), 8.65–8.70 (m, 2H), 11.50 (s, 4H, 2 NH_2_). ^13^C NMR δ: 55.83, 107.64, 110.62, 112.69 (2), 116.38, 117.35, 122.51, 129.16 (2), 131.18, 146.72, 148.30, 149.46, 150.97, 152.16, 158.52. Anal. Calcd. for C_17_H_17_N_5_O_3_S (371.11): C, 54.97; H, 4.61; N, 18.86. Found: C, 55.36; H, 5.00; N, 19.11.

#### 3.1.5. 4-((7-Methoxyquinolin-4-yl)amino)-N-(pyridin-2-yl)benzenesulfonamide **(3d)**

Yield 71%, m.p. 231–233 °C. IR: 3383, 3279 (2NH), 3078 (arom.), 2945, 2881 (aliph.), 1631 (CN), 1340, 1167 (SO_2_). ^1^H-NMR δ: 3.82 (s, 3H, OCH_3_), 6.50 (t, 1H, *J* = 5 Hz), 6.72 (d, 1H, *J* = 7 Hz), 7.15 (t, 1H, *J* = 5 Hz), 7.30–7.45 (m, 4H), 7.60 (d, 2H, *J =* 8 Hz), 7.78 (t, 1H, *J =* 7 Hz), 7.93 (d, 1H, *J =* 6 Hz), 8.07 (t, 1H, *J = 7* Hz), 8.41 (d, 1H, *J* = 7 Hz), 8.56 (s, 1H, NH), 10.30 (s, 1H, SO_2_NH). ^13^C NMR δ: 56.21, 101.98, 104.05, 133.89, 114.30 (2), 115.86, 118.54 (2), 122.43, 125.02, 128.76 (2), 137.69, 141.12, 143.27, 143.49, 145.83, 147.02, 150.43, 153.73. Anal. Calcd. for C_21_H_18_N_4_O_3_S (406.11): C, 62.05; H, 4.46; N, 13.78. Found: C, 61.92; H, 4.15; N, 13.40.

#### 3.1.6. 4-((7-Methoxyquinolin-4-yl)amino)-N-(pyrimidin-2-yl)benzenesulfonamide **(3e)**

Yield 66%, m.p. 188–190 °C. IR: 3325, 3232 (2NH), 3105 (arom.), 2950, 2923 (aliph.), 1630 (CN), 1357, 1132 (SO_2_). ^1^H-NMR δ: 3.86 (s, 3H, OCH_3_), 6.78 (d, 1H, J = 7 Hz), 6.93 (dd, 1H, J = 6 Hz), 7.20–7.30 (m, 4H), 7.79 (d, 2H, J = 7.5 Hz, AB), 7.95 (d, 1H, J = 8 Hz), 8.55–8.62 (m, 3H), 8.70 (s, 1H, NH), 10.72 (s, 1H, SO_2_NH). ^13^C NMR δ: 55.94, 105.46, 106.76, 112.61 (2), 115.96, 116.29, 119.28, 125.26, 126.26, 130.32 (2), 149.05, 149.80, 151.89 (2), 153.52, 158.72 (2), 160.40. Anal. Calcd. for C_20_H_17_N_5_O_3_S (407.11): C, 58.96; H, 4.21; N, 17.19. Found: C, 59.34; H, 4.52; N, 17.49.

#### 3.1.7. 4-((7-Methoxyquinolin-4-yl)amino)-N-(5-methylisoxazol-3-yl)benzenesulfonamide **(3f)**

Yield 70%, m.p. 92–94 °C. IR: 3302, 3234 (2NH), 3100 (arom.), 2959, 2868 (aliph.), 1618 (CN), 1378, 1147 (SO_2_). ^1^H-NMR δ: 2.51 (s, 3H, CH_3_), 3.87 (s, 3H, OCH_3_), 6.31 (s, 1H), 6.68 (d, 1H, *J* = 8.5 Hz), 7.23 (d, 2H, *J =* 8.5 Hz, AB), 7.34–7.42 (m, 2H), 7.69 (d, 2H, *J =* 8.5 Hz, AB), 8.05 (d, 1H, *J =* 6 Hz), 8.70–8.81 (m, 2H), 10.70 (s, 1H, SO_2_ NH). ^13^C NMR δ: 13.21, 56.14, 95.14, 108.22 (2), 119.83 (2), 120.93 (2), 121.17, 125.31 (3), 141.33 (2), 150.92 (2), 151.24 (2), 161.35. Anal. Calcd. for C_20_H_18_N_4_O_4_S (410.10): *C*, 58.53; H, 4.42; N, 13.65. Found: C, 58.86; H, 4.71; N, 13.97.

#### 3.1.8. 4-((7-Methoxyquinolin-4-yl)amino)-N-(thiazol-2-yl)benzenesulfonamide **(3g)**

Yield 71%, m.p. >300 °C. IR: 3346, 3360 (2NH), 3100 (arom.), 2920, 2846 (aliph.), 1617 (CN), 1360, 1136 (SO_2_). ^1^H-NMR δ: 3.79 (s, 3H, OCH_3_), 6.58 (d, 1H, *J* = 8.5 Hz), 6.81 (d, 1H, *J* = 9 Hz), 7.30–7.48 (m, 5H), 7.56 (d, 2H, *J =* 7 Hz, AB), 8.10 (d, 1H, *J =* 8.5 Hz), 8.55 (d, 1H, *J =* 8 Hz), 8.76 (s, 1H, NH), 10.30 (s, 1H, SO_2_NH). ^13^C NMR δ: 54.89, 103.07, 108.23, 112.94, 116.63 (2), 119.89, 121.22, 124.63, 129.01, 129.80 (2), 141.44, 142.16, 143.73, 151.29 (2), 153.79, 172.09. Anal. Calcd. for C_19_H_16_N_4_O_3_S_2_ (412.07): C, 55.32; H, 3.91; N, 13.58. Found: C, 55.68; H, 4.22; N, 13.89.

#### 3.1.9. 4-((7-Methoxyquinolin-4-yl)amino)-N-(4-methylpyrimidin-2-yl)benzenesulfonamide **(3h)**

Yield 73%, m.p. 200–202 °C. IR: 3346, 3310 (2NH), 3112 (arom.), 2925, 2856 (aliph.), 1619 (CN), 1366, 1146 (SO_2_). ^1^H-NMR δ: 2.34 (s, 3H, CH_3_), 3.82 (s, 3H, OCH_3_), 6.88 (d, 1H, *J* = 8.5 Hz), 6.90–7.03 (m, 4H), 7.75 (d, 2H, *J =* 8 Hz, AB), 8.15 (d, 1H, *J =* 6 Hz), 8.25–8.32 (m, 3H), 8.80 (s, 1H, NH), 10.76 (s, 1H, SO_2_NH). ^13^C NMR δ: 23.81, 56.50, 107.65, 112.46 (2), 114.06, 115.23, 118.88, 119.23, 125.26, 129.67, 130.53 (2), 142.17, 143.86, 153.47 (2), 157.35, 158.09 (2), 168.43. Anal. Calcd. for C_21_H_19_N_5_O_3_S (421.12): C, 59.84; H, 4.54; N, 16.62. Found: C, 60.13; H, 4.83; N, 16.99.

#### 3.1.10. N-(4,5-Dimethyloxazol-2-yl)-4-((7-methoxyquinolin-4-yl)amino)benzenesulfonamide **(3i)**

Yield 68%, m.p. 238–240 °C. IR: 3376, 3239 (2NH), 3096 (arom.), 2942, 2871 (aliph.), 1621 (CN), 1358, 1124 (SO_2_). ^1^H-NMR δ: 2.61 (s, 3H, CH_3_), 2.78 (s, 3H, CH_3_), 3.85 (s, 3H, OCH_3_), 6.70 (d, 1H, *J* = 7.5 Hz), 7.20–7.34 (m, 4H), 7.78 (d, 2H, *J* = 8 Hz, AB), 7.98 (d, 1H, *J* = 6.5 Hz), 8.39 (d, 1H, *J* = 7 Hz), 8.55 (d, 1H, *J* = 7 Hz), 9.49 (s, 1H, NH), 10.70 (s, 1H, SO_2_NH). ^13^C-NMR δ: 31.20, 36.23, 55.84, 105.31, 106.45, 112.65 (2), 116.90, 119.10, 125.60, 126.24, 129.35, 130.13 (2), 144.94, 145.05, 148.65, 149.48, 151.82, 153.38. Anal. Calcd. for C_21_H_20_N_4_O_4_S (424.12): C, 59.42; H, 4.75; N, 13.20. Found: C, 59.11; H, 4.41; N, 12.88.

#### 3.1.11. 4-((7-Methoxyquinolin-4-yl)amino)-N-(5-methyl-1,3,4-thiadiazol-2-yl)benzenesulfonamide **(3j)**

Yield 69%, m.p. 160–162 °C. IR: 3372, 3217 (2NH), 3100 (arom.), 2923, 2894 (aliph.), 1632 (CN), 1348, 1131 (SO_2_). ^1^H-NMR δ: 2.65 (s, 3H, CH_3_), 3.82 (s, 3H, OCH_3_), 6.60 (d, 1H, *J* = 8.5 Hz), 7.32–7.53 (m, 4H), 7.79 (d, 2H, *J =* 9 Hz, AB), 8.11 (d, 1H, *J =* 6 Hz), 8.32 (d, 1H, *J* = 9 Hz), 9.00 (s, 1H, NH), 10.70 (s, 1H, SO_2_NH). ^13^C-NMR δ: 16.46, 56.14, 106.54, 108.10, 113.07 (2), 114.79, 118.19, 120.99, 128.16 (2), 134.89, 142.08, 143.45, 145.38, 148.36, 149.37, 151.76, 168.67. Anal. Calcd. for C_19_H_17_N_5_O_3_S (427.08): C, 53.38; H, 4.01; N, 16.38. Found: C, 52.99; H, 3.69; N, 15.98.

#### 3.1.12. N-(2,6-Dimethylpyrimidin-4-yl)-4-((7-methoxyquinolin-4-yl)amino)benzenesulfonamide **(3k)**

Yield 66%, m.p. semisolid. IR: 3373, 3267, 3232 (NH, NH_2_), 3078 (arom.), 2956, 2870 (aliph.), 1653 (2C=O), 1365, 1199 (SO_2_). ^1^H-NMR δ: 2.44 (s, 3H, CH_3_), 2.61 (s, 3H, CH_3_), 3.93 (s, 3H, OCH_3_), 6.58 (d, 1H, *J* = 7 Hz), 6.77 (s, 1H), 7.24–7.33 (m, 4H), 7.77 (d, 2H, *J =* 6.5 Hz, AB), 8.05 (d, 1H, *J =* 8 Hz), 8.53 (d, 1H, *J = 8* Hz), 8.99 (s, 1H, NH), 10.67 (s, 1H, SO_2_NH). ^13^C-NMR δ: 16.35, 16.78, 56.16, 102.37, 105.34, 108.10, 114.79 (2), 118.35, 120.57, 121.73, 127.49, 128.16 (2), 145.38, 148.36, 149.37, 150.12, 152.90, 155.83, 161.76, 165.32. Anal. Calcd. for C_22_H_21_N_5_O_3_S (435.14): C, 60.67; H, 4.86; N, 16.08. Found: C, 61.03; H, 5.22; N, 16.41.

#### 3.1.13. N-(4,6-Dimethylpyrimidin-2-yl)-4-((7-methoxyquinolin-4-yl)amino)benzenesulfonamide **(3l)**

Yield 74%, m.p. 170–172 °C. IR: 3371, 3230 (2NH), 3095 (arom.), 2954, 2870 (aliph.), 1625 (CO), 1618 (CN), 1328, 1159 (SO_2_). ^1^H-NMR δ: 2.40 (s, 6H, 2CH_3_), 3.85 (s, 3H, OCH_3_), 6.36 (d, 1H, *J* = 7.5 Hz), 6.54 (s, 1H), 7.28 (d, 2H, *J =* 7 Hz, AB), 7.46–7.52 (m, 2H), 7.62 (d, 2H, *J =* 7 Hz, AB), 8.01 (d, 1H, *J =* 6 Hz), 8.62–8.65 (m, 2H), 11.20 (s, 1H, SO_2_NH). ^13^C NMR δ: 26.52 (2), 56.17, 107.86, 108.25 (2), 112.90 (2), 119.87, 120.96, 121.24, 125.38, 129.85 (2), 141.38, 150.94, 151.29 (2), 153.84, 164.84 (2), 172.05. MS (*m*/*z*, RI %): 435.16 (M^+^) (100). Anal. Calcd. for C_22_H_21_N_5_O_3_S (435.14): C, 60.67; H, 4.86; N, 16.08. Found: C, 60.91; H, 5.11; N, 16.38.

#### 3.1.14. N-(5-Methoxypyrimidin-2-yl)-4-((7-methoxyquinolin-4-yl)amino)benzenesulfonamide **(3m)**

Yield 72%, m.p. >300 °C. IR: 3362, 3267 (2NH), 3078 (arom.), 2962, 2868 (aliph.), 1618 (CN), 1363, 1134 (SO_2_). ^1^H-NMR δ: 3.85 (s, 6H, 2OCH_3_), 6.67 (d, 1H, *J* = 6.5 Hz), 7.30–7.55 (m, 4H), 7.70 (d, 2H, *J =* 3 Hz), 7.95 (d, 2H, *J =* 7 Hz, AB), 8.32–8.36 (m, 3H), 10.82 (s, 1H, SO_2_NH). ^13^C NMR δ: 55.84, 56.63, 105.31, 106.45, 112.65 (2), 116.90, 119.10, 125.60, 129.35, 130.13 (2), 144.94 (2), 145.05, 148.65, 149.48, 149.78, 151.82, 153.38, 162.79. Anal. Calcd. for C_21_H_19_N_5_O_4_S (437.12): C, 57.66; H, 4.38; N, 16.01. Found: C, 58.00; H, 4.62; N, 16.33.

#### 3.1.15. N-(6-Methoxypyrimidin-4-yl)-4-((7-methoxyquinolin-4-yl)amino)benzenesulfonamide **(3n)**

Yield 62%, m.p. >300 °C. IR: 3356, 3253 (2NH), 3104 (arom.), 2928, 2890 (aliph.), 1621 (CN), 1338, 1142 (SO_2_). ^1^H-NMR δ: 3.79 (s, 6H, 2OCH_3_), 5.98 (s, 1H), 6.68 (d, 1H, *J* = 6 Hz), 7.40–7.55 (m, 4H), 7.68 (d, 2H, *J = 7* Hz, AB), 7.98 (d, 1H, *J =* 6 Hz), 8.30–8.65 (m, 3H), 10.73 (s, 1H, SO_2_NH). ^13^C NMR δ: 54.20, 55.74, 91.08, 107.31, 110.45, 112.93 (2), 116.34, 119.30, 125.50, 129.48 (3), 144.72, 147.35, 149.18 (2), 153.40, 157.84, 167.39, 170.08. Anal. Calcd. for C_21_H_19_N_5_O_4_S (437.12): C, 57.66; H, 4.38; N, 16.01. Found: C, 57.32; H, 3.99; N, 15.86.

#### 3.1.16. N-(1H-Indazol-6-yl)-4-((7-methoxyquinolin-4-yl)amino)benzenesulfonamide **(3o)**

Yield 69%, m.p. 240–242 °C. IR: 3375, 3216 (2NH), 3095 (arom.), 2958, 2871 (aliph.), 1616 (CN), 1351, 1136 (SO_2_). ^1^H-NMR δ: 3.86 (s, 3H, OCH_3_), 6.45–6.55 (m, 2H), 6.80 (s, 1H), 7.32 (d, 1H, *J* = 3 Hz), 7.42–7.50 (m, 3H), 7.61–7.78 (m, 3H), 7.90 (d, 1H, *J* = 8 Hz), 8.30 (s, 1H), 8.55 (d, 1H, *J* = 7 Hz), 9.89 (s, 1H, NH), 10.73 (s, 1H, SO_2_NH). ^13^C NMR δ: 56.51, 90.21, 103.75, 110.68 (2), 112.28, 112.90 (2), 116.30, 117.42, 118.03, 122.08, 128.96, 129.70 (2), 134.52, 143.01, 144.72, 148.30 (2), 149.40 (2), 152.97. Anal. Calcd. for C_23_H_19_N_5_O_3_S (445.12): C, 62.01; H, 4.30; N, 15.72. Found: C, 62.40; H, 4.64; N, 15.99.

#### 3.1.17. 4-((7-Methoxyquinolin-4-yl)amino)-N-(quinoxalin-2-yl)benzenesulfonamide **(3p)**

Yield 78%, m.p. 220–222 °C. IR: 3381, 3244 (2NH), 3108 (arom.), 2987, 2875 (aliph.), 1627 (CN), 1347, 1125 (SO_2_). ^1^H-NMR δ: 3.84 (s, 3H, OCH_3_), 6.70 (d, 1H, *J* = 6 Hz), 7.20–7.25 (m, 3H), 7.32 (d, 1H, *J* = 2.5 Hz), 7.67 (t, 2H, *J =* 9 Hz), 7.75 (t, 2H, *J =* 9 Hz), 7.80 (d, 2H, *J =* 8 Hz, AB), 7.96 (d, 1H, *J =* 6.5 Hz), 8.50–8.61 (m, 3H), 10.70 (s, 1H, SO_2_NH). ^13^C NMR δ: 55.93, 108.28, 112.71 (3), 119.25, 121.64, 124.45, 126.21, 127.54, 129.14 (2), 130.60, 131.13 (2), 138.36, 139.08, 139.78, 146.73, 150.15, 151.85, 152.44, 153.87, 160.45. MS (*m*/*z*, RI %): 457.72 (M^+^) (51.00), 459.68 (M + 2) (42.50), 217.50 (100). Anal. Calcd. for C_24_H_19_N_5_O_3_S (457.12): C, 63.01; H, 4.19; N, 15.31. Found: C, 62.93; H, 3.92; N, 14.98.

#### 3.1.18. N-(2,6-Dimethoxypyrimidin-4-yl)-4-((7-methoxyquinolin-4-yl)amino) benzenesulfonamide **(3q)**

Yield 78%, m.p. 170–172 °C. IR: 3387, 3200 (2NH), 3110 (arom.), 2978, 2892 (aliph.), 1629 (CN), 1352, 1142 (SO_2_). ^1^H-NMR δ: 3.78 (s, 6H, 2OCH_3_), 3.83 (s, 3H, 3OCH_3_), 5.91 (s, 1H), 6.51 (d, 1H, *J* = 7.5 Hz), 7.50–7.62 (m, 4H), 7.98 (d, 2H, *J =* 7 Hz, AB), 8.21 (d, 1H, *J =* 6 Hz), 8.62–8.65 (m, 2H), 10.51 (s, 1H, SO_2_NH). ^13^C NMR δ: 54.18, 54.89, 56.17, 84.66, 101.43, 108.23, 112.90 (2), 116.61, 121.21, 124.53, 125.37, 129.85 (2), 147.72, 148.30, 151.28 (2), 153.84, 160.62, 164.84, 172.05. Anal. Calcd. for C_22_H_21_N_5_O_5_S (467.13): C, 56.52; H, 4.53; N, 14.98. Found: C, 56.26; H, 4.23; N, 14.72.

#### 3.1.19. N-(5,6-Dimethoxypyrimidin-4-yl)-4-((7-methoxyquinolin-4-yl)amino) benzenesulfonamide **(3r)**

Yield 76%, m.p. 103–105 °C. IR: 3398, 3363 (2NH), 3079(arom.), 2952, 2865 (aliph.), 1620 (CN), 1343, 1139 (SO_2_). ^1^H-NMR δ: 3.75 (s, 3H, OCH_3_), 3.86 (s, 6H, 2OCH_3_), 6.60 (d, 1H, *J* = 8 Hz), 7.30–7.49 (m, 4H), 7.99 (d, 2H, *J =* 8 Hz, AB), 8.11 (d, 1H, *J =* 6.5 Hz), 8.20 (d, 1H, *J* = 9 Hz), 8.60–8.70 (m, 2H), 11.12 (s, 1H, SO_2_NH). ^13^C NMR δ: 54.37, 55.95, 56.48, 107.68, 112.59, 119.20 (2), 119.82, 121.31, 124.71, 126.41, 127.11, 130.28 (2), 147.12, 148.32, 149.18, 151.02, 151.19, 152.89, 160.61, 161.80. Anal. Calcd. for C_22_H_21_N_5_O_5_S (467.13): C, 56.52; H, 4.53; N, 14.98. Found: C, 56.82; H, 4.58; N, 15.14.

#### 3.1.20. 4-((7-Methoxyquinolin-4-yl)amino)-N-(1-phenyl-1H-pyrazol-5-yl)benzenesulfonamide **(3s)**

Yield 74%, m.p. 220–222 °C. IR: 3262, 3276 (2NH), 3096 (arom.), 2920, 2868 (aliph.), 1640 (CN), 1354, 1163 (SO_2_). ^1^H-NMR δ: 3.85 (s, 3H, OCH_3_), 6.50 (d, 1H, *J* = 6.5 Hz), 6.72 (d, 1H, *J* = 7 Hz), 7.29–7.40 (m, 4H), 7.41–7.75 (m, 8H), 7.94 (d, 1H, *J =* 6.5 Hz), 8.30 (d, 1H, *J = 7* Hz), 8.89 (s, 1H, NH), 11.10 (s, 1H, SO_2_NH). ^13^C NMR δ: 56.60, 105.50, 107.61, 110.49, 112.46 (2), 115.46 (2), 119.23 (3), 125.26, 129.68 (2), 130.31 (3), 136.80, 138.94, 141.31, 145.52 (2), 150.17, 151.07, 152.35. Anal. Calcd. for C_25_H_21_N_5_O_3_S (471.14): C, 63.68; H, 4.49; N, 14.85. Found: C, 64.00; H, 4.71; N, 15.04.

### 3.2. Antimicrobial Activity

The antimicrobial activity of the synthesized samples was evaluated using the agar well diffusion method on six microorganisms, including Gram-negative bacteria (*Escherichia coli* ATCC 25922 and *Pseudomonas aeruginosa* ATCC 27853), Gram-positive bacteria (*Staphylococcus aureus* ATCC 25923 and *Bacillus subtilis* ATCC 6051), and unicellular fungi (*Candida albicans* ATCC 90028 and *Cryptococcus neoformans* ATCC 14116). The pathogenic microbes that were examined were known to cause UTIs. While the fungal strains were inoculated on malt extract agar (MEA) dishes and incubated for 3–5 days at 28 ± 2 °C before being stored at 4 °C for further use, the tested bacteria were inoculated on nutrient agar for one day at 37 °C [72]. The minimal inhibitory concentration (MIC) of the synthesized samples was also established using the microdilution method. To establish the MIC, tests at various concentrations (ranging from 1000 to 0.5 µg/mL) for each substance were conducted. To evaluate the antimicrobial potential of the synthesized samples, the ZOI test must also be performed with amoxicillin/clavulanic acid (AMC), a common antibacterial drug, and nystatin (NS), a common antifungal positive control [61].

### 3.3. Antibiofilm Potential

Furthermore, a qualitative analysis concerning biofilm restraint was conducted, as described by Christensen et al. [73]. The definitive study of the biofilm, which was displayed on the tube wall, verified the absence and proximity of the biofilm in the integrated samples. The antibiofilm of the synthesized samples (at 10.0 µg/mL) was assessed against the selected microbes, tested, and correlated with the reference (non-treated one). The examined bacteria and fungi were then inoculated into 5 mL of the nutrient broth medium, which was then adjusted by 0.5 McFarland to be 1–3.5 × 10^8^ CFU/mL. Later, they were incubated at 37.0 ± 0.5 °C for 24 h. The media found in control and treated tubes were dropped, combined with Phosphate Buffer Saline (PBS; pH 7.0), and ultimately preserved. Next, the bacterial and yeast cells that adhered to the tube walls were implanted with 5 mL of sodium acetate (3.5%) for approximately 20 min. Finally, they were cleaned with de-ionized water. Biofilms organized inside tubes were stained with 20 mL of crystal violet (CV; 0.15%) and washed with de-ionized water to eliminate the CV. It must be noted that, for the semi-quantitative antibiofilm calculation, 5 mL of absolute ethanol was injected to separate the stained bacterial and yeast biofilms [74]. A UV-Vis. spectrophotometer at 570.0 nm measured the O.D. of the stained bacterial and yeast biofilms [60]. The bacterial and yeast biofilm hindrance percentage was determined by using the subsequent relation (Equation (1)) [75]:Biofilm inhibition % = (O.D. Control sample − O.D. treated sample)/O.D. Control sample × 100(1)

### 3.4. Growth Curve Assay

The growth curve assay determined the influence of compound **3l** on the growth of *E. coli* (the most sensitive microbes), according to the method of Huang et al. [76]. The bacterial suspension was adjusted to 0.5 McFarland (1 × 10^8^ CFU/mL) in 5.0 mL nutrient broth tubes. Compound **3l** was included separately in every examined tube. The absorbance of the bacterial growth following treatment was evaluated at 2 h time intervals up to 24 h (wavelength of 600 nm) [77]. To obtain the regular growth curve, the average of duplicate measurements was compared to those of the hourly intervals.

### 3.5. Effect of Compound **3l** on Protein Leakage from Bacterial Cell Membranes

A pure 18 h bacterial culture was set at 0.5 McFarland (1 × 10^8^ CFU/mL) and 100 µL was injected into 10 mL of the nutrient broth, including compound **3l**. As a control, a broth without compound **3l** was infused with culture. All of the treated samples were centrifuged for 15 min at 5000 rpm after being kept at 37 °C for 5 h [78]. For the different samples, 100 μL of supernatants were combined with 1 mL of Bradford reagent. Optical density was measured at 595 nm after 10 min of dark incubation [78].

### 3.6. Reaction Mechanism Using SEM Analysis

The sensitive microbial cells (*E. coli*) were washed with PBS three times and fixed with a 4.0% glutaraldehyde solution [79]. The preserved microbial cells were regularly cleaned with PBS and repeatedly drained with various ethanol concentrations (30, 50, 70, 90, and 100%) for 15 min at 28 ± 2 °C [80]. The fixed samples were then solidified on a piece of aluminum for SEM determination. SEM analysis was used to examine the morphological traits of the control (non-treated microbial cell) and treated microbes.

### 3.7. Physicochemical and Pharmacokinetic Parameters

All physicochemical parameters and pharmacokinetics were calculated using swissADME, a free online web tool, (http://www.swissadme.ch/, accessed on 21 April 2023).

### 3.8. Statistical Analysis

The statistical analysis of the obtained results was performed after applying the ONE-WAY ANOVA (at *p* < 0.05) and Duncan’s methods [81]. The accepted findings were examined using SPSS software version 15.

## 4. Conclusions

A new series of quinoline derivatives bearing sulfonamide **3(a–s)**, was synthesized from the starting material, 4-chloro-7-methoxyquinoline **1**. The target compounds were designed and synthesized to be evaluated as antimicrobial agents for various Gram-positive bacteria, Gram-negative bacteria, and fungi. Compound **3l** was the most potent in this study. Overall, all the designed compounds exhibited promising antimicrobial activity against the tested bacterial and fungal strains compared to AMC/Nyst as conventional antimicrobial agents, where compounds **3c**, **3d**, and **3l** were significantly more active than AMC/Nyst. The antibiofilm results showed that the highest percentage of inhibition after the addition of 10.0 µg/mL of compound **3l** was 94.60% for *E. coli*, 91.74% for *P. aeruginosa,* and 98.03% for *C. neoformans*. Note that compound **3l** could achieve biofilm extension at its adhesion strength, which is the initial start in the antimicrobial method. In the growth curve assay, the control sample’s *E. coli* development rate appeared to be rapid. The O.D. of the control sample at λ = 600 nm was 2.18. In contrast, the OD_600_ value of the compound **3l**-treated cells was lower than that of the control sample due to the exceptional antibacterial action of compound **3l**. In the protein leakage assay, it was observed that the quantity of cellular protein discharged from *E. coli* was directly proportional to the concentration of compound **3l,** which was determined to be 180.25 µg/mL after the treatment with 1.0 mg/mL of compound **3l**. This proved the antibacterial characteristics of the synthesized compound **3l**, and explained the creation of holes in the cell membrane of *E. coli*, which produce the oozing out of the proteins from the *E. coli* cytoplasm. The physicochemical and pharmacokinetic properties of compounds **3c**, **3d**, and **3l** were also estimated to determine their drug-like properties. According to the obtained results, the newly targeted compounds are regarded as promising scaffolds for the continued development of novel antimicrobials.

## Data Availability

All data generated or analyzed for this study are included in this published paper (and its Appendix A).

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
