# Peer review of "Synthesis, Antimicrobial, and Antibiofilm Activities of Some Novel 7-Methoxyquinoline Derivatives Bearing Sulfonamide Moiety against Urinary Tract Infection-Causing Pathogenic Microbes"

_ijms, 2023, doi:10.3390/ijms24108933_

Round 1

Reviewer 1 Report

The paper has addressed the synthesis, antimicrobial and antibiofilm activities of sulfonamide derivatives against bacterial and fungal strains that cause urinary tract infections.

The research topic addressed in this paper is relevant to the field. The authors themselves have previously published research findings on synthesized compound derivatives. The literature gap has been mentioned in the introduction as well as comparison of previous work with present work has also been explained.

In this paper, the synthesis of a new series of 4-((7-methoxyquinolin-4-yl) amino)-N-(substituted) benzenesulfonamide has been reported via the reaction of 4-chloro-7-methoxyquinoline with various sulfa drugs. The paper has presented antimicrobial activity, antibiofilm potential, growth curve assay, SEM and statistical analysis of synthesized compounds. The synthesized compounds have also been characterized via 1H and 13C NMR spectroscopy.

In Scheme 1.. a number should be assigned to sulfonamide reactant. 

Structure-Activity-Relationship studies should be added in the manuscript. 

To add more value to the manuscript, authors are suggested to add related docking/in-silico studies.

 In characterization of compounds, mass values are missing which need to be added. Supplementary information should be provided in the form of a separate file containing all the spectra of the synthesized molecules.

Minor remarks:

In the 5th line of abstract, “that” should be placed after “revealed”.

Line 7 of abstract.. add a space after equal sign.

Line 11 of abstract.. Italicize E. coli

Line 1 of introduction.. word “use” has been repeated. Rephrase the sentence

Line 1 of introduction.. “Results”

Reference 1 isn’t relevant

Line 2 of introduction.. bacteria strains should be replaced by bacterial strains

Line 3 of introduction.. Replace making infections with causing infections

Line 3 of page 2.. use more appropriate word than quickly

Line 6 of page 2.. italicize bacterial name

Page 3. Line 8.. towards

Last paragraph of page 3. Line 5.. omit antimicrobial

Page 5..a number should be assigned to sulfonamide reactant.

Compound number should be bold throughout the paper

Page 7.. subheading 2.3.. The first three lines should be rephrased for better understanding

Page 7.. first line of second last paragraph should not be bold.

Page 15.. subheading 4.4.. line 2.. remove extra spacing.

English language needs to be improved and typos should be removed.

Author Response

Ms. Ref. No.:  IJMS-2360752

Title: Synthesis, antimicrobial and antibiofilm activities of some novel 7-methoxyquinoline derivatives bearing sulfonamide moiety against urinary tract infection-causing pathogenic microbes.

Dear Editor-In-Chief;

Thank you for your email enclosing the reviewer’s comments. We have carefully reviewed the comments and are pleased to submit the revised version of the MS. Following the reviewer’s advice, we have addressed each of their concerns in a point-by-point manner below and also highlighted them in the MS.

Response to Reviewer# 1:

In Scheme 1. a number should be assigned to sulfonamide reactant. 

Response: Sulfonamide reactants were assigned numbers.

Structure-Activity-Relationship studies should be added in the manuscript. 

Response: SAR study was added to the MS.

To add more value to the manuscript, authors are suggested to add related docking/in-silico studies.

Response: In-Silico ADMET study was performed and added to the MS, based on the reviewer’s comment.

 In characterization of compounds, mass values are missing which need to be added. Supplementary information should be provided in the form of a separate file containing all the spectra of the synthesized molecules.

Response: Mass spectra were performed for some compounds and were added to the supplementary file. Also, the Supplementary data file was uploaded as a separate file.

Minor remarks:

In the 5th line of abstract, “that” should be placed after “revealed”.

Response: Corrected

Line 7 of abstract.. add a space after equal sign.

Response: Corrected

Line 11 of abstract.. Italicize E. coli.

Response: Corrected

Line 1 of introduction.. word “use” has been repeated. Rephrase the sentence.

Response: Corrected

Line 1 of introduction.. “Results”.

Response: Corrected

Reference 1 isn’t relevant.

Response: Corrected

Line 2 of introduction.. bacteria strains should be replaced by bacterial strains

Response: Corrected

Line 3 of introduction.. Replace making infections with causing infections

Response: Corrected

Line 3 of page 2.. use more appropriate word than quickly

Response: Corrected

Line 6 of page 2.. italicize bacterial name

Response: Corrected

Page 3. Line 8.. towards

Response: Corrected

Last paragraph of page 3. Line 5.. omit antimicrobial

Response: Corrected

Page 5..a number should be assigned to sulfonamide reactant.

Response: Corrected

Compound number should be bold throughout the paper.

Response: Corrected

Page 7.. subheading 2.3.. The first three lines should be rephrased for better understanding

Response: Corrected

Page 7.. first line of second last paragraph should not be bold.

Response: Corrected

Page 15.. subheading 4.4.. line 2.. remove extra spacing.

Response: Corrected

English language needs to be improved and typos should be removed.

Response: Corrected

Reviewer 2 Report

Overall the manuscript is well-written and the experiments are well-executed. The English language in some cases is difficult to understand, and the author needs to language edit their manuscript.

The author has studied on the synthesis, antimicrobial and antibiofilm activities of some of the novel 7-methoxyquinoline derivatives bearing sulfonamide moiety against the urinary tract infection-causing pathogenic microorganisms. The current article is suitable for the journal and it addresses an important topic. Overall the manuscript is well-written and the experiments are well-executed. However, the English language in some cases is difficult to understand, and the author needs to language edit their manuscript. Besides, some grammatical errors such as writing the sentences in past tense and converting the scientific names in italic is required. All the compound names in the main text can be made bold so that it can be easy to identify by the readers while going through the whole manuscript. The authors can provide all the spectral figures (1H and 13C NMR ) of the identified compounds as supporting materials.

 The English language in some cases is difficult to understand, and the author needs to language edit their manuscript.

Author Response

Ms. Ref. No.:  IJMS-2360752

Title: Synthesis, antimicrobial and antibiofilm activities of some novel 7-methoxyquinoline derivatives bearing sulfonamide moiety against urinary tract infection-causing pathogenic microbes.

Dear Editor-In-Chief;

Thank you for your email enclosing the reviewer’s comments. We have carefully reviewed the comments and are pleased to submit the revised version of the MS. Following the reviewer’s advice, we have addressed each of their concerns in a point-by-point manner below and also highlighted them in the MS.

Response to reviewer# 2:

Overall the manuscript is well-written and the experiments are well-executed. The English language in some cases is difficult to understand, and the author needs to language edit their manuscript.

Response: A language edit was performed.

- write the pathogens' names in Italic

Response: Done

- rephrase the sentence, it doesn't have logic: "Quinoline rings form the core of many drugs acting as antivi-rals including anti SARS-COV-2 [25, 26], antimalarial [27], antituberculosis [28], antipara-sitic [29] and anticancer agents [30]."

Response: Rephrased.

- rephrase the sentence, it doesn't have logic: "While the oxazole derivative 10 exhibited 2CH3 groups that showed two singlets at 2.11 & 2.28 ppm and signals at 31.20 & 36.23 ppm."

Response: Rephrased.

- when you present the results obtained, you should use the past tense.

Response: Done

- bold compounds 4 and 5 in page 6 and compound 13 in page 7

Response: Done.

Reviewer 3 Report

Dear Authors,

The subject your manuscript addresses is important for the medicinal chemists, especially because of the microbial resistance to current therapy used in the treatment of urinary infections.

I have some suggestions for improving the quality of your paper:

- write the pathogens' names in Italic

- rephrase the sentence, it doesn't have logic: "Quinoline rings form the core of many drugs acting as antivi-rals including anti SARS-COV-2 [25, 26], antimalarial [27], antituberculosis [28], antipara-sitic [29] and anticancer agents [30]."

- rephrase the sentence, it doesn't have logic: "While the oxazole derivative 10 exhibited 2CH3 groups that showed two singlets at 2.11 & 2.28 ppm and signals at 31.20 & 36.23 ppm."

- when you present the results obtained, you should use the past tense

- bold compounds 4 and 5 in page 6 and compound 13 in page 7

One important aspect that is missing from your manuscript and that is needed for a better scientific value is a SAR study (structure-activity relationship). In medicinal chemistry, it is not enough to identify the more potent compound, but it is required to analyze why, which are the structural fragments responsible for the antimicrobial effect. You say that your compounds are hybrids, therefore you should investigate what are the roles of the two pharmacophores: quinoline and sulfonamide.

Also, there is needed an ADMET study for the compounds synthesized.

The quality of English Language has to be improved, there are phrases without verbs and no logic, in consequence.

Author Response

Ms. Ref. No.:  IJMS-2360752

Title: Synthesis, antimicrobial and antibiofilm activities of some novel 7-methoxyquinoline derivatives bearing sulfonamide moiety against urinary tract infection-causing pathogenic microbes.

Dear Editor-In-Chief;

Thank you for your email enclosing the reviewer’s comments. We have carefully reviewed the comments and are pleased to submit the revised version of the MS. Following the reviewer’s advice, we have addressed each of their concerns in a point-by-point manner below and also highlighted them in the MS.

Response to reviewer# 3:

One important aspect that is missing from your manuscript and that is needed for a better scientific value is a SAR study (structure-activity relationship). In medicinal chemistry, it is not enough to identify the more potent compound, but it is required to analyze why, which are the structural fragments responsible for the antimicrobial effect. You say that your compounds are hybrids, therefore you should investigate what are the roles of the two pharmacophores: quinoline and sulfonamide.

Also, there is needed an ADMET study for the compounds synthesized.

 Response: SAR study and in silico ADMET study were performed.

The quality of English Language has to be improved, there are phrases without verbs and no logic, in consequence.

Response: Corrected

Round 2

Reviewer 1 Report

The authors have now made the desired changes to the manuscript which can now be accepted in current form.

Overall, English language is fine.

Author Response

This is the reviewer comment:

The authors have now made the desired changes to the manuscript which can now be accepted in current form.

Overall, English language is fine.

Reviewer 3 Report

Dear Authors,

I am glad that I could help improving the scientific aspect of your manuscript. You made some corrections needed, but still there are some aspects that aren't solved:

1. you added a SAR discussion, but in my opinion, the information presented is not quite a SAR study. All you have done is the enumeration of the activity displayed by each compound. You have to discuss the relationship between the chemical structures and the activity. Therefore, it is needed to investigate what are the fragments that act as pharmacophores, why some act better than others (fragments that are hydrophils or lipophils, electron donors, etc.)

 2. you say that you added the ADME study, but I do not find it in the manuscript

Author Response

Response to Reviewer# 3:

I am glad that I could help improving the scientific aspect of your manuscript. You made some corrections needed, but still there are some aspects that aren't solved:

  1. you added a SAR discussion, but in my opinion, the information presented is not quite a SAR study. All you have done is the enumeration of the activity displayed by each compound. You have to discuss the relationship between the chemical structures and the activity. Therefore, it is needed to investigate what are the fragments that act as pharmacophores, why some act better than others (fragments that are hydrophils or lipophils, electron donors, etc.)

Response: Thanks to the reviewer’s comments, the SAR study was improved.

  1. you say that you added the ADME study, but I do not find it in the manuscript.

Response: ADME study was highlighted

Round 3

Reviewer 3 Report

Dear Authors,

Thank you for considering all my suggestions made in order to improve the quality of your manuscript.